# Bayesian and non-bayesian inference for logistic-exponential distribution using improved adaptive type-II progressively censored data

**Subhankar Dutta**[1], **Hana N. Alqifari**[2]*, **Amani Almohaimeed**[2]

**1** Division of Mathematics, School of Advanced Sciences, Vellore Institute of Technology, Chennai, India,
**2** Department of Statistics and Operation Research, College of Science, Qassim University, Buraydah, Saudi Arabia

* hn.alqifari@qu.edu.sa

## Abstract

Improved adaptive type-II progressive censoring schemes (IAT-II PCS) are increasingly being used to estimate parameters and reliability characteristics of lifetime distributions, leading to more accurate and reliable estimates. The logistic exponential distribution (LED), a flexible distribution with five hazard rate forms, is employed in several fields, including lifetime, financial, and environmental data. This research aims to enhance the accuracy and reliability estimation capabilities for the logistic exponential distribution under IAT-II PCS. By developing novel statistical inference methods, we can better understand the behavior of failure times, allow for more accurate decision-making, and improve the overall reliability of the model. In this research, we consider both classical and Bayesian techniques. The classical technique involves constructing maximum likelihood estimators of the model parameters and their asymptotic covariance matrix, followed by estimating the distribution's reliability using survival and hazard functions. The delta approach is used to create estimated confidence intervals for the model parameters. In the Bayesian technique, prior information about the LED parameters is used to estimate the posterior distribution of the parameters, which is derived using Bayes' theorem. The model's reliability is determined by computing the posterior predictive distribution of the survival or hazard functions. Extensive simulation studies and real-data applications assess the effectiveness of the proposed methods and evaluate their performance against existing methods.

## 1 Introduction

Reliability analysis has important implications across several fields, encompassing engineering, health, and finance, for assuring system safety and performance. Dealing with censored data highlights one of the challenges in reliability analysis since it occurs when a component's failure time is not fully observed, either because the component has not yet failed or because the experiment has been terminated before the component failed. Traditional censoring systems,

**Data Availability Statement:** All relevant data are within the manuscript.

**Funding:** Researchers would like to thank the Deanship of Scientific Research, Qassim University for funding publication of this project.

**Competing interests:** The authors have declared that no competing interests exist.

**Abbreviations:** ACIs, Approximate confidence intervals; AT-II PCS, adaptive progressive Type-II censoring scheme; CSs, Censoring schemes; IAT-II PCS, improved adaptive progressive Type-II censoring scheme; LE, Logistic exponential; LED, Logistic exponential distribution; PCS, progressive censoring scheme; Type-II PHCS, progressive Type-II hybrid censoring scheme.

such as Type-I, Type-II, and hybrid schemes, have been thoroughly investigated, however they lack flexibility in eliminating units at any time. To address this limitation, Cohen [1] introduced the progressive censoring scheme (PCS). In PCS, units are extracted from the experiment at different time points, with the number of units extracted at each time point specified in advance. This allows for more experiment design flexibility, improved efficiency for certain research questions, and a reduction in evaluate the duration and expenses to a specific extent in comparison to conventional CSs. PCSs have been used in a variety of research settings, including the study of product lifetimes, the reliability of engineering systems, and the survival of patients with diseases. They offer several advantages over conventional CSs, including increased flexibility and efficiency. For more details on PCS, one may refer to [2–5]. However, PCSs can be more complex to design and analyze, and they may not be suitable for all research questions.

The increased reliability of products resulting from technological advances in manufacturing can lengthen experimental periods in accordance with progressive type-II censoring schemes. In order to address this limitation, Kundu and Joarder [6] proposed the type-II progressive hybrid censoring scheme. Considerable research has been devoted to the examination of this particular scheme ([7–9]). Nevertheless, in the context of this censoring scheme, the experimental duration is fixed, resulting in a randomization of the effective sample size. There are situations in which the effective sample size can reach zero, hence diminishing the efficacy of statistical inference. In order to address this inefficiency, Ng et al. [10] introduced the adaptive type-II progressive censoring scheme (AT-II PCS), a combination of Type-II progressive censoring and Type-I censoring schemes. This scheme comes to an end once a predetermined quantity of failure-time data has been obtained. For more details on this scheme, one may refer to [11–16]. Nonetheless, this experiment could take an exceptionally long time to complete if the experimental devices are exceptionally dependable. In order to address this limitation, Yan et al. [17] introduced the IAT-II PCS, which is an enhanced adaptive type-II progressive censored scheme. By modifying the AT-II PCS, this strategy circumvents the problem of lengthy experimental periods. The IAT-II PCS efficiently guarantees that tests conclude within a predetermined timeframe. Hence, the suggested IAT-II PCS can be employed when studies need to be concluded within a designated timeframe. In statistical literature, IAT-II PCS has been explored by a few researchers (see [18, 19]). There is still ample opportunity for further investigation on the topic of parameter estimation for various distributions using IAT-II PCS.

Lan and Leemis [20] introduced the logistic exponential distribution (LED). The probability density function (PDF) and cumulative distribution function (CDF) of LED are defined respectively, as

$$f(x) = \frac{\nu\eta \ e^{\nu x}(e^{\nu x} - 1)^{\eta-1}}{\left(1 + (e^{\nu x} - 1)^{\eta}\right)^2} \ ; \nu, \eta > 0, \ \ x > 0, \tag{1}$$

and

$$F(x) = P(X \leq x) = 1 - \frac{1}{1 + (e^{\nu x} - 1)^{\eta}}. \tag{2}$$

Accordingly, the survival function (SF) is given by

$$S(x) = P(X > x) = \frac{1}{1 + (e^{\nu x} - 1)^{\eta}} \ ; \ \ x > 0. \tag{3}$$

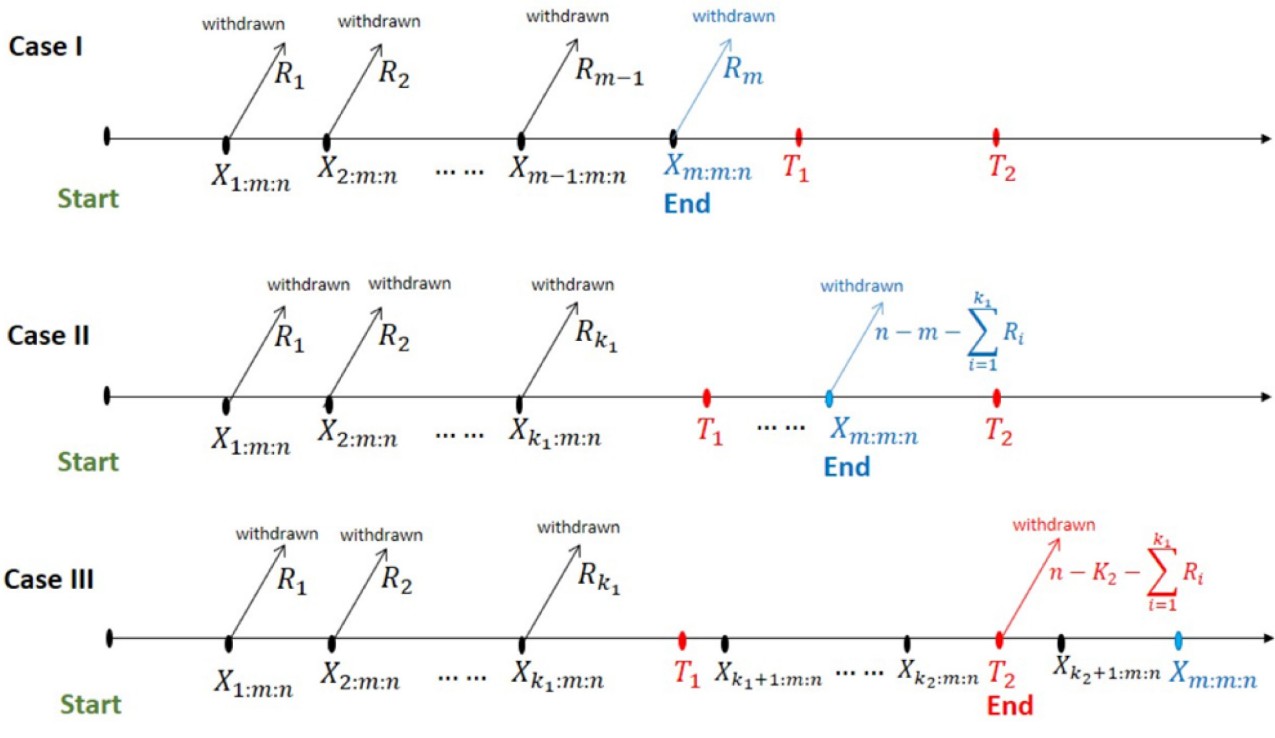

**Fig 1. PDF and HRF plots of the LED for different values of $v$ and $\eta$.**

The LED displays a wide variety of hazard rate shapes, including constant, rising, decreasing, bathtub, and upside-down bathtub. The hazard rate function (HRF) is defined as follows:

$$h(t) = \frac{v\eta \, e^{vt}(e^{vt} - 1)^{\eta-1}}{1 + (e^{vt} - 1)^{\eta}} \; ; \quad t > 0. \tag{4}$$

The plots of PDF and HRF of the LED for different values of the parameters have been depicted in Fig 1. For values of $\eta$ less than 1, the hazard rate function exhibits an increasing trend. Conversely, for values of $\eta$ greater than 1, the hazard rate function shows a decreasing trend. Finally, when $\eta$ equals 1, the hazard rate function remains constant. Furthermore, in the case when $\eta < 1$ and $v$ is of small magnitude, the hazard rate function exhibits a bathtub shape. Conversely, when $\eta > 1$ and $v$ is of significant magnitude, it has an inverted bathtub form. It should be noted that variously shaped hazard rate functions are used with varying meanings in reliability engineering, biology, and various statistical modelings. Consider a high failure rate in infancy that rises with age, then falls to a certain extent, stays constant for a while, and then rises again. A bathtub-shaped hazard rate model may be used to depict this kind of scenario. In the early stages of a functioning system, the decreasing hazard rate is often seen. Such hazard model may suit the earthquake data set. Furthermore, since increasing hazard rate models have consequences for the assessment of certain objective functions used to describe stochastic occurrences, they are employed in research related to operations and supply chain management. In biology, it is shown throughout the course of a disease whose mortality peaks after a certain amount of time and then steadily decreases. Thus, an upside-down bathtub-shaped hazard rate model may be used to represent the associated data set. LED has all these types of hazard rate. Owing to the many forms of the hazard rate function, the LED exhibits great

adaptability in several research domains, including clinical, reliability, and survival investigations. This work specifically examines censored samples using an enhanced adaptive Type-II progressive censoring method. The objective is to estimate the unknown parameters of the LED.

As far as we know, we have not seen any research on the estimate of model parameters and reliability features of the LED under IAT-II PCS. Therefore, this work aims to address and eliminate this disparity. Initially, the parameters, SF and HRF of the LED are estimated using the frequentist estimation methodology, namely the traditional maximum likelihood (MLE) method. Additionally, the asymptotic confidence intervals (ACIs) are computed the LED parameters. The second aim is to acquire the Bayesian estimate (BEs) of the unknown LED parameters using gamma priors and the accompanying greatest posterior density credible intervals. Due to the unavailability of closed equations for the BEs, Markov chain Monte Carlo (MCMC) techniques are employed to compute the intricate posterior functions. This allows for the calculation of the BEs and the corresponding highest posterior density (HPD) credible intervals. The performance of the suggested approaches is evaluated using a comprehensive Monte Carlo simulation analysis, focusing on indicators such as mean square error (MSE), average length, coverage probability, and average bias. Data obtained from real-world engineering applications is also subjected to analysis.

The structure of the paper is as follows: Section 2 introduces the adaptive type-II progressive censoring scheme (AT-II PCS). Section 3 focuses on several frequentist approaches for estimating the parameters of the LED model. The Bayes estimators are discussed in Section 4. Section 5 showcases the Monte Carlo simulation to compare the proposed estimates. A real-life data set is used in Section 6 to demonstrate the practical use of the LED using IAT-II PCS in real-world phenomena. Finally, Section 7 ends up with some concluding remarks.

## 2 Model formulation

IAT-II PCS emerges in a reliability investigation in the following manner: Given an assumption of a distribution and a probability density function denoted as $f(x)$, as well as a cumulative distribution function denoted as $F(x)$, an experiment starts with a set of $n$ indistinguishable objects. The experiment started with a PCS $R = (R_1, R_2, \ldots, R_{m-1}, R_m = n - m - \sum_{i=1}^{m-1} R_i)$, where $R_i \geq 0$, and the predetermined failure number $m$ ($m < n$). The results of the experiment's performance can be used to modify the value of $R_i$. Simultaneously, there are two predetermined timing thresholds, denoted as $T_1$ and $T_2$, where ($T_1, T_2 \in (0, \infty)$) and $T_1 < T_2$. According to the IAT-II PCS, there are three observed cases as follows:

**Case I**: When $X_{m:m:n} < T_1$, the associated sample is denoted as $X_{1:m:n}, X_{2:m:n}, \cdots, X_{m:m:n}$, which refers to type-II PCS.

**Case II**: When $T_1 < X_{m:m:n} < T_2$, the related sample is denoted as $X_{1:m:n}, X_{2:m:n}, \cdots, X_{m:m:n}$, which refers to AT-II PCS.

**Case III**: When $T_2 < X_{m:m:n}$, subsequently, the termination time becomes $T_2$ and $X_{k_2:m:n} < T_2 \leq X_{k_2+1:m:n} \leq X_{m:m:n}$. The IAT-II PCS sample in this instance is represented as $X_{1:m:n}, \cdots, X_{k_2:m:n}$. The experiment conducted using the IAT-II PCS is concluded when time $T^*$ is equal to the minimum value between $X_{m:m:n}$ and $T_2$. Fig 2 illustrates the three scenarios visually.

If an experiment contains $n$ units following a random variable $X$ with CDF $F(x; \theta)$ and PDF $f(x; \theta)$ with a PCS $R = R_1, \cdots, R_{m-1}, R_m = n - m - \sum_{i=1}^{m-1} R_i$. Then, according to [17], the

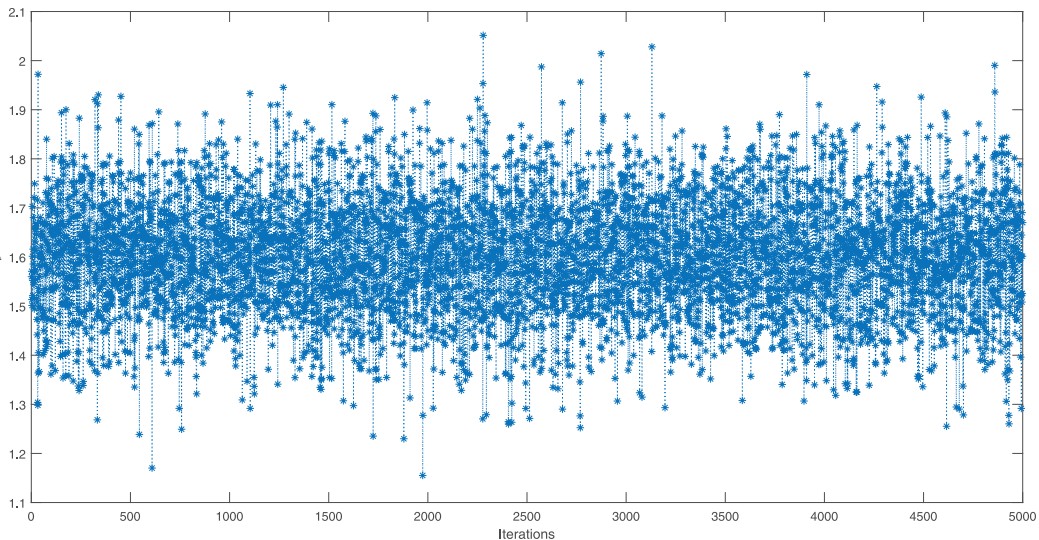

**Fig 2. Schematic representation of the improved adaptive type-II progressive censored scheme.**

likelihood function using IAT-II PCS is given below:

$$L(v, \eta|\mathbf{x}) = C \left[ \prod_{i=1}^{E_2} f(x_{i:m:n}) \right] \left[ \prod_{i=1}^{E_1} (1 - F(x_{i:m:n}))^{R_i} \right] [1 - F(T^*)]^B, \tag{5}$$

where $C$ is a constant that is independent of the parameters and the quantities $E_2$, $E_1$, $R_i$, $T^*$, and $B$ for the various situations are shown in Table 1. The failure counts prior to $T_2$ and $T_1$ are denoted by $E_2$ and $E_1$, respectively.

## 3 Classical estimation

This section develops the classical estimates of the parameters and the reliability characteristics of the LED based on IAT-II PCS.

**Table 1. Diverse selections of the quantities $E_2$, $E_1$, $C$, $T^*$, and $B$.**

|  | $E_2$ | $E_1$ | $C$ | $T^*$ | $B$ |
|---|---|---|---|---|---|
| Case I | $m$ | $m$ | $\prod_{i=1}^{m}(n - i + 1 - \sum_{s=1}^{i-1} R_s)$ | $X_{m:m:n}$ | $B_1 = 0$ |
| Case II | $m$ | $k_1$ | $\prod_{i=1}^{m}(n - i + 1 - \sum_{s=1}^{k_1} R_s)$ | $X_{m:m:n}$ | $B_2 = n - m - \sum_{i=1}^{k_1} R_i$ |
| Case III | $k_2$ | $k_1$ | $\prod_{i=1}^{k_2}(n - i + 1 - \sum_{s=1}^{i-1} R_s)$ | $T_2$ | $B_3 = n - k_2 - \sum_{i=1}^{k_1} R_i.$ |

### 3.1 Maximum likelihood estimation

When $X$ follows LED, using (1), (2) and (5), the likelihood function based on IAT-II PCS is stated as follows:

$$
\begin{aligned}
L(v,\eta|\mathbf{x}) \quad &= C\left[\prod_{i=1}^{E_2}\frac{v\eta(e^{vx_i}-1)^{\eta-1}e^{vx_i}}{(1+(e^{vx_i}-1)^\eta)^2}\right]\left[\prod_{i=1}^{E_1}\left(\frac{1}{1+(e^{vx_i}-1)^\eta}\right)^{R_i}\right]e^{-W}\\
&= C\left[\prod_{i=1}^{E_2}\frac{v\eta\Psi_{x_i}^{\eta-1}(\Psi_{x_i}+1)}{(1+\Psi_{x_i}^\eta)^2}\right)\right]\left[\prod_{i=1}^{E_1}(1+\Psi_{x_i}^\eta)^{-R_i}\right]e^{-W},
\end{aligned}
\tag{6}
$$

where $x_i = x_{i:m:n}$, $\Psi_{x_i} = (e^{vx_i}-1)$, and

$$
W = \begin{cases}
0 & \text{Case I,}\\
B_2 ln(1+\Psi_{x_m}^\eta) & \text{Case II,}\\
B_3 ln(1+\Psi_{T_2}^\eta) & \text{Case III.}
\end{cases}
\tag{7}
$$

The log-likelihood function,

$$
\begin{aligned}
l(v,\eta|\mathbf{x}) \quad &\propto E_2 ln(v\eta) + (\eta-1)\sum_{i=1}^{E_2}ln\Psi_{x_i} + \sum_{i=1}^{E_2}vx_i - 2\sum_{i=1}^{E_2}ln(1+\Psi_{x_i}^\eta)\\
&-\sum_{i=1}^{E_1}R_i ln(1+\Psi_{x_i}^\eta) - W.
\end{aligned}
$$

The log-likelihood $l(v,\eta|\mathbf{x})$ has the following derivatives of first order with regard to $v$ and $\eta$:

$$
\begin{aligned}
\frac{\partial l(v,\eta|\mathbf{x})}{\partial v} \quad &= \frac{E_2}{v} + (\eta-1)\sum_{i=1}^{E_2}\frac{x_i e^{vx_i}}{\Psi_{x_i}} + \sum_{i=1}^{E_2}x_i - 2\eta\sum_{i=1}^{E_2}\frac{x_i\Psi_{x_i}^{\eta-1}e^{vx_i}}{(1+\Psi_{x_i}^\eta)}\\
&-\eta\sum_{i=1}^{E_1}R_i\frac{x_i\Psi_{x_i}^{\eta-1}e^{vx_i}}{(1+\Psi_{x_i}^\eta)} - W'^{(v)},
\end{aligned}
\tag{8}
$$

where

$$
W'^{(v)} = \begin{cases}
0 & \text{Case I,}\\
B_2\eta\dfrac{x_m\Psi_{x_m}^{\eta-1}e^{vx_m}}{(1+\Psi_{x_m}^\eta)} & \text{Case II,}\\
B_3\eta\dfrac{T_2\Psi_{T_2}^{\eta-1}e^{vT_2}}{(1+\Psi_{T_2}^\eta)} & \text{Case III,}
\end{cases}
$$

and

$$
\frac{\partial l(v,\eta|\mathbf{x})}{\partial \eta} = \frac{E_2}{\eta} + \sum_{i=1}^{E_2}ln\Psi_{x_i} - 2\sum_{i=1}^{E_2}\frac{\Psi_{x_i}^\eta ln\Psi_{x_i}}{1+\Psi_{x_i}^\eta} - \sum_{i=1}^{E_1}R_i\frac{\Psi_{x_i}^\eta ln\Psi_{x_i}}{1+\Psi_{x_i}^\eta} - W'^{(\eta)},
\tag{9}
$$

where

$$W'^{(\eta)} = \begin{cases} 0 & \text{Case I,} \\[2ex] B_2 \dfrac{\Psi_{x_m}^{\eta} ln\Psi_{x_m}}{1+\Psi_{x_m}^{\eta}} & \text{Case II,} \\[3ex] B_3 \dfrac{\Psi_{T_2}^{\eta} ln\Psi_{T_2}}{1+\Psi_{T_2}^{\eta}} & \text{Case III.} \end{cases}$$

Setting (8) and (9) equal to zero:

$$\frac{E_2}{v} + (\eta-1)\sum_{i=1}^{E_2}\frac{x_i e^{vx_i}}{\Psi_{x_i}} + \sum_{i=1}^{E_2}x_i - 2\eta\sum_{i=1}^{E_2}\frac{x_i\Psi_{x_i}^{\eta-1}e^{vx_i}}{(1+\Psi_{x_i}^{\eta})} - \eta\sum_{i=1}^{E_1}R_i\frac{x_i\Psi_{x_i}^{\eta-1}e^{vx_i}}{(1+\Psi_{x_i}^{\eta})} - W'^{(v)} = 0, \qquad (10)$$

and

$$\frac{E_2}{\eta} + \sum_{i=1}^{E_2}ln\Psi_{x_i} - 2\sum_{i=1}^{E_2}\frac{\Psi_{x_i}^{\eta}ln\Psi_{x_i}}{1+\Psi_{x_i}^{\eta}} - \sum_{i=1}^{E_1}R_i\frac{\Psi_{x_i}^{\eta}ln\Psi_{x_i}}{1+\Psi_{x_i}^{\eta}} - W'^{(\eta)} = 0. \qquad (11)$$

The system of nonlinear equations represented by Eqs (10) and (11) lacks a closed-form solution due to the intractable nature of the terms involved. Consequently, numerical methods are indispensable for obtaining the ML estimates $\hat{v}_{MLE}$ and $\hat{\eta}_{MLE}$. The Newton-Raphson method is a common numerical approach employed to find these estimates.

Substituting the MLEs $\hat{v}_{MLE}$ and $\hat{\eta}_{MLE}$ into formula (3) and (4), then the MLEs of reliability characteristics have been acquired as

$$\hat{S}(t)_M = \frac{1}{1+\left(e^{\hat{v}_{MLE}t}-1\right)^{\hat{\eta}_{MLE}}},$$

and

$$\hat{h}(t)_M = \frac{\hat{v}_{MLE}\hat{\eta}_{MLE}\left(e^{\hat{v}_{MLE}t}-1\right)^{\hat{\eta}_{MLE}-1}e^{\hat{v}_{MLE}t}}{1+\left(e^{\hat{v}_{MLE}t}-1\right)^{\hat{\eta}_{MLE}}}.$$

## 3.2 Asymptotic confidence interval

In this section, the ACIs for the model parameters and the reliability characteristics have been constructed based on the normality properties of the MLEs of the parameters.

**3.2.1 Confidence intervals of parameters.** Based on the asymptotic normality of the relevant MLEs, the ACIs of $v$, $\eta$, $S(t)$, and $h(t)$ are determined. To obtain the ACIs, the asymptotic variance-covariance matrix (VCM) has to be constructed based on the MLEs. This VCM can be obtained from the inverse of the Fisher information matrix (FIM), and the FIM is expressed below:

$$I(v,\eta) = E\begin{bmatrix} -\dfrac{\partial^2 l(v,\eta|\mathbf{x})}{\partial v^2} & -\dfrac{\partial^2 l(v,\eta|\mathbf{x})}{\partial v\partial\eta} \\[3ex] -\dfrac{\partial^2 l(v,\eta|\mathbf{x})}{\partial\eta\partial v} & -\dfrac{\partial^2 l(v,\eta|\mathbf{x})}{\partial\eta^2} \end{bmatrix}. \qquad (12)$$

Here,

$$\frac{\partial^2 l(v,\eta|\mathbf{x})}{\partial v^2} = -\frac{E_2}{v^2} - (\eta-1)\sum_{i=1}^{E_2}\frac{x_i^2 e^{vx_i}}{\Psi_{x_i}^2} - 2\eta\sum_{i=1}^{E_2}z_{x_i} - \eta\sum_{i=1}^{E_1}R_i z_{x_i} - W''^{(v)},$$

$$z_{x_i} = \frac{x_i^2\Psi_{x_i}^{\eta-2}e^{vx_i}(\eta e^{vx_i}-\Psi_{x_i}^{\eta}-1)}{(1+\Psi_{x_i}^{\eta})^2},$$

where

$$W''(v) = \begin{cases} 0 & \text{Case I,} \\ B_2\eta z_{x_m} & \text{Case II,} \\ B_3\eta z_{T_2} & \text{Case III,} \end{cases}$$

$$\frac{\partial^2 l(v,\eta|\mathbf{x})}{\partial v\partial\eta} = \sum_{i=1}^{E_2}\frac{x_i e^{vx_i}}{\Psi_{x_i}} - 2\sum_{i=1}^{E_2}x_i e^{vx_i}\frac{\Psi_{x_i}^{\eta-1}(ln(\Psi_{x_i})\eta+\Psi_{x_i}^{\eta}+1)}{(1+\Psi_{x_i}^{\eta})^2}$$
$$-\sum_{i=1}^{E_1}R_i x_i e^{vx_i}\frac{\Psi_{x_i}^{\eta-1}(ln(\Psi_{x_i})\eta+\Psi_{x_i}^{\eta}+1)}{(1+\Psi_{x_i}^{\eta})^2} - W'^{(v,\eta)},$$

$$W'(v,\eta) = W'(\eta,v) = \begin{cases} 0 & \text{Case I,} \\ B_2 x_m e^{vx_m}\dfrac{\Psi_{x_m}^{\eta-1}(ln(\Psi_{x_m})\eta+\Psi_{x_m}^{\eta}+1)}{(1+\Psi_{x_m}^{\eta})^2} & \text{Case II,} \\ B_3 T_2 e^{vT_2}\dfrac{\Psi_{T_2}^{\eta-1}(ln(\Psi_{T_2})\eta+\Psi_{T_2}^{\eta}+1)}{(1+\Psi_{T_2}^{\eta})^2} & \text{Case III,} \end{cases}$$

$$\frac{\partial^2 l(v,\eta|\mathbf{x})}{\partial\eta\partial v} = \sum_{i=1}^{E_2}\frac{x_i e^{vx_i}}{\Psi_{x_i}} - 2\sum_{i=1}^{E_2}\frac{x_i\Psi_{x_i}^{\eta-1}e^{vx_i}(\eta ln\Psi_{x_i}+\Psi_{x_i}^{\eta}+1)}{(1+\Psi_{x_i}^{\eta})^2}$$
$$-\sum_{i=1}^{E_1}R_i\frac{x_i\Psi_{x_i}^{\eta-1}e^{vx_i}(\eta ln\Psi_{x_i}+\Psi_{x_i}^{\eta}+1)}{(1+\Psi_{x_i}^{\eta})^2} - W'(\eta,v),$$

and

$$\frac{\partial^2 l(v,\eta|\mathbf{x})}{\partial\eta^2} = -\frac{E_2}{\eta^2} - 2\sum_{i=1}^{E_2}\frac{\Psi_{x_i}^{\eta}}{(1+\Psi_{x_i}^{\eta})^2}ln\Psi_{x_i}^2 - \sum_{i=1}^{E_1}R_i\frac{\Psi_{x_i}^{\eta}}{(1+\Psi_{x_i}^{\eta})^2}ln\Psi_{x_i}^2 - W''^{(\eta)},$$

where

$$
W''(\eta) = \begin{cases} 0 & \text{Case I,} \\[2ex] B_2 \dfrac{\Psi_{x_m}^{\eta}}{\left(1 + \Psi_{x_m}^{\eta}\right)^2} ln\Psi_{x_m}^2 & \text{Case II,} \\[3ex] B_3 \dfrac{\Psi_{T_2}^{\eta}}{\left(1 + \Psi_{T_2}^{\eta}\right)^2} ln\Psi_{T_2}^2 & \text{Case III.} \end{cases}
$$

Finding the precise expressions for the aforementioned expectation (12) is challenging. Therefore, by using the observed FIM $\hat{I}(\hat{v}_{MLE}, \hat{\eta}_{MLE})$, one can easily obtain the approximate FIM as

$$
I_{OF}(\hat{v}_{MLE}, \hat{\eta}_{MLE})) = \begin{bmatrix} -\dfrac{\partial^2 l(v, \eta|X)}{\partial v^2} & -\dfrac{\partial^2 l(v, \eta|X)}{\partial v \partial \eta} \\[3ex] -\dfrac{\partial^2 l(v, \eta|X)}{\partial \eta \partial v} & -\dfrac{\partial^2 l(v, \eta|X)}{\partial \eta^2} \end{bmatrix}_{(v,\eta)=(\hat{v}_{MLE}, \hat{\eta}_{MLE})}
$$

Furthermore, based on the asymptotic normality properties of the MLEs,

$$
(\hat{v}_{MLE}, \hat{\eta}_{MLE}) \sim N((v, \eta), I_{OF}^{-1}(\hat{v}_{MLE}, \hat{\eta}_{MLE}))),
$$

where

$$
I_{OF}^{-1}(\hat{v}_{MLE}, \hat{\eta}_{MLE}) = \begin{bmatrix} var(\hat{v}_{MLE}) & cov(\hat{v}_{MLE}, \hat{\eta}_{MLE}) \\[2ex] cov(\hat{\eta}_{MLE}, \hat{v}_{MLE}) & var(\hat{\eta}_{MLE}) \end{bmatrix}
$$

For the parameters $v$ and $\eta$, the corresponding $100(1 - \xi)\%$ ACIs are given as

$$
\left[ \hat{v}_{MLE} - z_{\xi/2}\sqrt{var(\hat{v}_{MLE})}, \hat{v}_{MLE} + z_{\xi/2}\sqrt{var(\hat{v}_{MLE})} \right]
$$

$$
\left[ \hat{\eta}_{MLE} - z_{\xi/2}\sqrt{var(\hat{\eta}_{MLE})}, \hat{\eta}_{MLE} + z_{\xi/2}\sqrt{var(\hat{\eta}_{MLE})} \right]
$$

where $z_{\xi/2}$ is the upper percentile of $N(0, 1)$.

**3.2.2 ACIs of $S(t)$ and $h(t)$.** Since the functions of the ML estimators are intractable when attempting to calculate the variance analytically, the Delta method (see [21, 22]) is utilized to evaluate the approximate confidence intervals for the survival functions $S(t)$. The survival function is then approximated linearly, and the variance of the linear approximation is subsequently determined as follows:

let $G_1^T = \left[\frac{\partial S(t)}{\partial v}, \frac{\partial S(t)}{\partial \eta}\right]$ and $G_2^T = \left[\frac{\partial h(t)}{\partial v}, \frac{\partial h(t)}{\partial \eta}\right]$ where, $G_l^T$ represents the transpose of $G_l$, and

$$\frac{\partial S(t)}{\partial v} = \frac{t\eta(e^{vt}-1)e^{vt}}{(1+(e^{vt}-1)^\eta)^2} = \frac{t\eta\Psi_t e^{vt}}{(1+\Psi_t^\eta)^2}$$

$$\frac{\partial S(t)}{\partial \eta} = \frac{(e^{vt}-1)^\eta ln(e^{vt}-1)}{(1+(e^{vt}-1)^\eta)^2} = \frac{\Psi_t^\eta ln\Psi_t}{(1+\Psi_t^\eta)^2}$$

$$\frac{\partial h(t)}{\partial v} = \frac{\eta(e^{vt}-1)^{\eta-2}e^{vt}[((e^{vt}-1)^\eta+\eta vt+1)e^{vt}+(-tv-1)(e^{vt}-1)^\eta-tv-1]}{(1+(e^{vt}-1)^\eta)^2}$$

$$\frac{\partial h(t)}{\partial \eta} = \frac{v(e^{vt}-1)^{\eta-1}e^{vt}[ln(e^{vt}-1)\eta+(e^{vt}-1)^\eta+1]}{(1+(e^{vt}-1)^\eta)^2}$$

The delta method yields the following approximations for the asymptotic variances of $\hat{S}_{MLE}(t)$ and $\hat{h}_{MLE}(t)$:

$$var(\hat{S}_{MLE}(t)) \simeq \left[G_1^T I_{OF}^{-1} G_1\right]|_{\hat{v}_{MLE},\hat{\eta}_{MLE}}$$

and

$$var(\hat{h}_{MLE}(t)) \simeq \left[G_2^T I_{OF}^{-1} G_2\right]|_{\hat{v}_{MLE},\hat{\eta}_{MLE}}$$

As a result, $S(t)$ and $h(t)$ have the following ACIs:

$$\left[\hat{S}_{MLE}(t) - z_{\xi/2}\sqrt{var(\hat{S}_{MLE}(t))}, \hat{S}_{MLE}(t) + z_{\xi/2}\sqrt{var(\hat{S}_{MLE}(t))}\right]$$

and

$$\left[\hat{h}_{MLE}(t) - z_{\xi/2}\sqrt{var(\hat{h}_{MLE}(t))}, \hat{h}_{MLE}(t) + z_{\xi/2}\sqrt{var(\hat{h}_{MLE}(t))}\right]$$

## 4 Bayesian estimation

The earlier sections concentrated on using frequentist methods to estimate the LED model's parameters. However, the main focus of this section will be on estimating $v$ and $\eta$, as well as $S(t)$ and $h(t)$, of the LED using Bayesian approaches. The ability to modify the support of the LED distribution makes gamma priors more flexible than other forms of priors. Furthermore, independent gamma priors are not very complicated, thus they should not lead to any computing issues or complicated posterior expressions. Specifically, the parameters $v$ and $\eta$ are assumed to follow independent gamma prior distributions with probability density functions

$G(a_1, b_1)$ and $G(a_2, b_2)$, respectively, as indicated by the following equations:

$$\pi_1(v; a_1, b_1) = \frac{b_1^{a_1}}{\Gamma(a_1)} v^{a_1-1} e^{-b_1 v}; v > 0, a_1, b_1 > 0,$$

$$\pi_2(\eta; a_2, b_2) = \frac{b_2^{a_2}}{\Gamma(a_2)} \eta^{a_2-1} e^{-b_2 \eta}; \eta > 0, a_2, b_2 > 0.$$

Hence, the joint prior distribution is as follows:

$$\pi(v, \eta) \propto \eta^{a_2-1} \; v^{a_1-1} \; e^{-(b_2\eta + b_1 v)}.$$

Based on the likelihood function in (6), the joint posterior density function of $v$ and $\eta$ is as follows,

$$\pi(v, \eta | \mathbf{x}) = \frac{\pi(v, \eta) \; L(v, \eta | \mathbf{x}) \; dv d\eta}{\int_0^\infty \int_0^\infty \pi(v, \eta) \; L(v, \eta | \mathbf{x}) \; dv d\eta} \tag{13}$$

By multiplying the complete data likelihood by the joint prior and reducing the result by a normalizing constant called the marginal likelihood, one can derive the Bayesian estimates of any function of $v$ and $\eta$, represented by $\psi(v, \eta)$, under squared error loss function (SELF) and LINEX loss function (LLF) which are defined as

$$\pi(\delta, \hat{\delta}) = (\delta - \hat{\delta})^2, \tag{14}$$

and

$$\psi(\delta, \hat{\delta}) = e^{p(\hat{\delta} - \delta)} - \eta(\hat{\delta} - \delta) - 1, \; p \neq 0, \tag{15}$$

respectively and $\hat{\delta}$ is the estimator of $\delta$. Then the Bayes estimates under SELF and LLF can be obtained through the following formula:

$$\hat{\psi}_{SE}(v, \eta) = \frac{\int_0^\infty \int_0^\infty \psi(v, \eta) \pi(v, \eta) \; L(v, \eta | \mathbf{x}) \; dv d\eta}{\int_0^\infty \int_0^\infty \pi(v, \eta) \; L(v, \eta | \mathbf{x}) \; dv d\eta}, \tag{16}$$

and

$$\hat{\psi}_{LI}(v, \eta) = -\left(\frac{1}{p}\right) \frac{\int_0^\infty \int_0^\infty e^{-p\psi(v,\eta)} \pi(v, \eta) \; L(v, \eta | \mathbf{x})}{\int_0^\infty \int_0^\infty \pi(v, \eta) \; L(v, \eta | \mathbf{x}) \; dv d\eta}, \; p \neq 0. \tag{17}$$

From the above expressions in (16) and (17), the Bayes estimates can not be obtained explicitly because of having ratio of two integrals. Estimating the marginal likelihood, represented by the term in the denominator, is a difficult issue, often involving a high-dimensional integration of the likelihood over the prior distribution. Nevertheless, Markov Chain Monte Carlo (MCMC) methods offer a solution to this problem by generating samples from the joint posterior distribution without requiring the marginal likelihood to be explicitly estimated.

## 4.1 MCMC method

MCMC method uses the joint posterior distribution (13) in terms of

$$\pi(v, \eta | \mathbf{x}) \propto v^{a_1-1} \; \eta^{a_2-1} \left[ \prod_{i=1}^{E_2} \frac{v \Psi_{x_i}^{\eta-1}(\Psi_{x_i} + 1)}{(1 + \Psi_{x_i}^{\eta})^2} \right] \left[ \prod_{i=1}^{E_1} (1 + \Psi_{x_i}^{\eta})^{-R_i} \right] e^{-W-(b_2\eta + b_1 v)}$$

The conditional distributions for $v$ and $\eta$ can be obtained as

$$\pi_1(v|\eta, \mathbf{x}) \propto v^{a_1+E_2-1} \left[\prod_{i=1}^{E_2} v(\Psi_{x_i}+1)\right] e^{-b_1 v},$$

and

$$\pi_2(\eta|v, \mathbf{x}) \propto \eta^{a_2-1} \left[\prod_{i=1}^{E_2} \frac{\Psi_{x_i}^{\eta-1}}{(1+\Psi_{x_i}^{\eta})^2}\right] \left[\prod_{i=1}^{E_1} (1+\Psi_{x_i}^{\eta})^{-R_i}\right] e^{-W-b_2 \eta}$$

These conditional distributions can not be expressed in a known distributional form. So in this case the Metropolis-Hastings (M-H) algorithm will be used to generate the MCMC samples for the parameters and corresponding reliability characteristics. The samples have been generated by using the following algorithm:

Algorithm 1

**Step 1:** Set $k = 1$ and $(v^{(0)}, \eta^{(0)}) = (\hat{v}, \hat{\eta})$ as initial values of $v$ and $\eta$.

**Step 2:** Generate $v^{(j)}$ and $\eta^{(j)}$ with normal distribution as $v^{(k)} \sim N(v^{(k-1)}, var(\hat{v}))$ and $\eta^{(k)} \sim N(\eta^{(k-1)}, var(\hat{\eta}))$.

**Step 3:** Compute $\Lambda_v = \min\left(1, \frac{\pi_1(v^{(k)}|\mathbf{x}, \eta^{(k-1)})}{\pi_2(v^{(k-1)}|\mathbf{x}, \eta^{(k-1)})}\right)$ and $\Lambda_\eta = \min\left(1, \frac{\pi_1(\eta^{(k)}|\mathbf{x}, v^{(k-1)})}{\pi_1(\eta^{(k-1)}|\mathbf{x}, v^{(k-1)})}\right)$.

**Step 4:** Generate samples for $\phi_1$ and $\phi_2$ from Uniform (0, 1) distribution.

**Step 5:** Set

$$\begin{cases} v^{(k)} = v^{(k)}, & \text{if } \phi_1 \leq \Lambda_v; \text{ otherwise } v^{(k)} = v^{(k-1)}, \\ \eta^{(k)} = \eta^{(k)}, & \text{if } \phi_2 \leq \Lambda_\eta; \text{ otherwise } \eta^{(k)} = \eta^{(k-1)} \end{cases}$$

**Step 6:** Set $k = k + 1$.

**Step 7:** Repeat **Step 2** to **Step 6**, P times to get $\{v^{(1)}, \cdots, v^{(P)}\}$ and $\{\eta^{(1)}, \cdots, \eta^{(P)}\}$.

**Step 8:** Using these samples, samples of the reliability characteristics $\{S^{(1)}(t), \cdots, S^{(P)}(t)\}$ and $\{h^{(1)}(t), \cdots, h^{(P)}(t)\}$.

Thus, using these MCMC samples the Bayes estimates of $v$, $\eta$, $S(t)$, and $h(t)$ under SELF and LLF functions can be obtained as

$$\hat{v}_{SEL} = \sum_{k=Q+1}^{P} \frac{v^{(k)}}{P-Q}, \quad \hat{\eta}_{SEL} = \sum_{k=Q+1}^{P} \frac{\eta^{(k)}}{P-Q},$$

$$\hat{S}_{SEL}(t) = \sum_{k=Q+1}^{P} \frac{S^{(k)}(t)}{P-Q}, \quad \hat{h}_{SEL}(t) = \sum_{k=Q+1}^{P} \frac{h^{(k)}(t)}{P-Q},$$

$$\hat{v}_{LL} = -\left(\frac{1}{p}\right) \log\left[\frac{\sum_{k=Q+1}^{P} e^{-pv^{(k)}}}{P-Q}\right], \quad \hat{\eta}_{LL} = -\left(\frac{1}{p}\right) \log\left[\frac{\sum_{k=Q+1}^{P} e^{-p\eta^{(k)}}}{P-Q}\right],$$

$$\hat{S}_{LL}(t) = -\left(\frac{1}{p}\right) \log\left[\frac{\sum_{k=Q+1}^{P} e^{-pS^{(k)}(t)}}{P-Q}\right], \quad \hat{h}_{LL}(t) = -\left(\frac{1}{p}\right) \log\left[\frac{\sum_{k=Q+1}^{P} e^{-ph^{(k)}(t)}}{P-Q}\right].$$

respectively, where $Q$ is the burn-in period. Further, to construct the HPD credible intervals of $(v, \eta, S(t), h(t))$, arrange the samples $(v^{(k)}, \eta^{(k)}, S^{(k)}(t), h^{(k)}(t))$ for $k = 1, \cdots, P$, in an increasing order to get $(v^{[k]}, \eta^{[k]}, S^{[k]}(t), h^{[k]}(t))$ for $k = 1, \cdots, P$. Then the $100(1 - \xi)\%$ HPD credible

intervals of $v$, $\eta$, $S(t)$, and $h(t)$ are respectively given by

$$\left(v^{[(P-Q)\xi/2]}, v^{[(P-Q)(1-\xi/2)]}\right), \quad \left(\eta^{[(P-Q)\xi/2]}, \eta^{[(P-Q)(1-\xi/2)]}\right),$$
$$\left(S^{[(P-Q)\xi/2]}(t), S^{[(P-Q)(1-\xi/2)]}(t)\right), \quad \text{and} \quad \left(h^{[(P-Q)\xi/2]}(t), h^{[(P-Q)(1-\xi/2)]}(t)\right).$$

## 5 Simulation study

A meticulous Monte Carlo simulation analysis is conducted to assess the behavior of the theoretical conclusions produced in the preceding sections, covering the classical and BEs and the corresponding credible and confidence intervals. The performance of the point estimates is compared based on average bias (AB) and mean squared error (MSE) and interval estimates have been compared based on the average width (AW) of the intervals and the coverage probability (CP). To achieve this, we have generated 10, 000 IAT-II PCS samples considering $v = 1.5$ and $\eta = 1.2$ using an algorithm developed by Yan et al. [17]. To estimate the reliability characteristics, set $t = 0.25$ for which $S(t) = 0.7201$ and $h(t) = 1.6111$. R 4.0.4, a type of programming interface, was used for all numerical calculations.

To conduct the simulation, several combinations of $n$, $m$, $T_1$, and $T_2$ have been taken into account with various censoring schemes. Two distinct possibilities of $n$ and $m$ are utilized for each given time $(T_1, T_2) = (0.4, 0.8)$ and $(0.55, 1.1)$, where $n = (40, 80)$ and $m$ is assumed to equal $(50, 75)\%$ for each $n$. The following three different PCS have been considered to remove the experimental units randomly at the time of the experiment:

- **Scheme I**: $R_1 = (n - m)$, $R_i = 0$, for $i = 2, \cdots, m$.

- **Scheme II**: $R_m = (n - m)$, $R_i = 0$, for $i = 1, \cdots, m - 1$.

- **Scheme III**: $R_{m/2} = (n - m)$, $R_i = 0$, for $i = 1, \cdots, m/2 - 1, m/2 + 1, \cdots, m$.

In the classical paradigm, MLEs and the 95% ACIs are obtained. In the Bayesian framework, the values of the hyperparameters $(u_i, v_i)$, for $i = 1, 2$ have been considered using the derivation given by Dutta and Kayal [23]. Classical estimates are better than BEs in situations when prior knowledge about the unknown parameters is lacking, as BEs need more computational expenses. To obtain Bayes estimates under LLF, we consider $p = 0.5$. To obtain Bayes estimates, the M-H algorithm has been used to generate MCMC samples. In this study, 10, 000 MCMC samples have been generated considering a 1, 000 burn-in period. Using these samples, the 95% HPD credible intervals are also constructed. Table 2 summarizes the ABs and MSEs of the point estimates. Whereas Table 3 represents the AWs and CPs of the interval estimates. The following are the key findings from Tables 2 and 3:

- **Effect of increasing** $m$: For fixed values of $n$, $T_1$ and $T_2$, increasing $m$ results in a decrease in the ABs and MSEs of the point estimates for $v$, $\eta$, $S(t)$, and $h(t)$. Additionally, the AWs of the ACI and HPD credible intervals decrease, while the CPs increase.

- **Effect of increasing** $n$: For fixed values of $m$, $T_1$ and $T_2$, increasing $n$ leads to a decrease in the ABs and MSEs of the point estimates for $v$, $\eta$, $S(t)$, and $h(t)$. Similar to the effect of increasing $m$, the AWs of the ACI and HPD credible intervals decrease, while the CPs increase.

- **Effect of increasing** $(T_1, T_2)$: For fixed values of $n$ and $m$, increasing $(T_1, T_2)$ results in a decrease in the ABs and MSEs of the point estimates for $v$, $\eta$, $S(t)$, and $h(t)$. However, the AWs of the ACI and HPD credible intervals decrease, while the CPs increase.

**Table 2. ABs and MSEs (in parenthesis) of the estimates of $v$, $\eta$, $S(0.25)$, and $h(0.25)$ with different values of $n$, $m$, and $(T_1, T_2)$.**

| $(n, m)$ | CS | | $T_1 = 0.4, T_2 = 0.8$ | | | $T_1 = 0.55, T_2 = 1.1$ | | |
|---|---|---|---|---|---|---|---|---|
| | | | **MLE** | **SELF** | **LLF** | **MLE** | **SELF** | **LLF** |
| (40,20) | I | $v$ | 0.1072 (0.0820) | 0.0673 (0.0445) | 0.0640 (0.0429) | 0.1011 (0.0796) | 0.0625 (0.0422) | 0.0596 (0.0413) |
| | | $\eta$ | 0.0988 (0.0790) | 0.0639 (0.0410) | 0.0604 (0.0398) | 0.0957 (0.0741) | 0.0617 (0.0395) | 0.0582 (0.0390) |
| | | $S(0.25)$ | 0.0136 (0.0095) | 0.0092 (0.0066) | 0.0090 (0.0065) | 0.0127 (0.0092) | 0.0090 (0.0065) | 0.0089 (0.0064) |
| | | $h(0.25)$ | 0.1752 (0.1051) | 0.1123 (0.0865) | 0.1086 (0.0849) | 0.1691 (0.0973) | 0.1098 (0.0837) | 0.1063 (0.0837) |
| | II | $v$ | 0.1105 (0.0846) | 0.0695 (0.0457) | 0.0668 (0.0441) | 0.1046 (0.0821) | 0.0649 (0.0437) | 0.0627 (0.0421) |
| | | $\eta$ | 0.1020 (0.0822) | 0.0652 (0.0421) | 0.0617 (0.0413) | 0.0984 (0.0752) | 0.0629 (0.0410) | 0.0603 (0.0399) |
| | | $S(0.25)$ | 0.0145 (0.0098) | 0.0094 (0.0068) | 0.0092 (0.0066) | 0.0131 (0.0093) | 0.0092 (0.0066) | 0.0091 (0.0065) |
| | | $h(0.25)$ | 0.1820 (0.1104) | 0.1196 (0.0894) | 0.1135 (0.0877) | 0.1735 (0.1031) | 0.1120 (0.0851) | 0.1099 (0.0846) |
| | III | $v$ | 0.1023 (0.0809) | 0.0658 (0.0433) | 0.0629 (0.0420) | 0.0995 (0.0785) | 0.0620 (0.0417) | 0.0592 (0.0410) |
| | | $\eta$ | 0.0972 (0.0761) | 0.0627 (0.0403) | 0.0598 (0.0395) | 0.0943 (0.0729) | 0.0612 (0.0390) | 0.0579 (0.0383) |
| | | $S(0.25)$ | 0.0131 (0.0092) | 0.0090 (0.0065) | 0.0089 (0.0064) | 0.0123 (0.0091) | 0.0089 (0.0064) | 0.0088 (0.0064) |
| | | $h(0.25)$ | 0.1716 (0.1032) | 0.1089 (0.0827) | 0.1047 (0.0815) | 0.1654 (0.0955) | 0.1064 (0.0828) | 0.1040 (0.0822) |
| (40,30) | I | $v$ | 0.0935 (0.0756) | 0.0622 (0.0406) | 0.0613 (0.0398) | 0.0910 (0.0741) | 0.0610 (0.0399) | 0.0582 (0.0392) |
| | | $\eta$ | 0.0920 (0.0731) | 0.0604 (0.0385) | 0.0591 (0.0376) | 0.0899 (0.0720) | 0.0598 (0.0378) | 0.0570 (0.0368) |
| | | $S(0.25)$ | 0.0124 (0.0090) | 0.0086 (0.0059) | 0.0082 (0.0056) | 0.0117 (0.0086) | 0.0085 (0.0058) | 0.0080 (0.0055) |
| | | $h(0.25)$ | 0.1589 (0.0972) | 0.1040 (0.0833) | 0.1015 (0.0821) | 0.1524 (0.0945) | 0.1012 (0.0824) | 0.0985 (0.0799) |
| | II | $v$ | 0.1061 (0.0823) | 0.0682 (0.0448) | 0.0657 (0.0438) | 0.1009 (0.0813) | 0.0636 (0.0428) | 0.0620 (0.0418) |
| | | $\eta$ | 0.0989 (0.0792) | 0.0643 (0.0411) | 0.0605 (0.0404) | 0.0976 (0.0741) | 0.0618 (0.0401) | 0.0592 (0.0393) |
| | | $S(0.25)$ | 0.0134 (0.0092) | 0.0090 (0.0066) | 0.0089 (0.0064) | 0.0122 (0.0091) | 0.0089 (0.0064) | 0.0088 (0.0063) |
| | | $h(0.25)$ | 0.1734 (0.1065) | 0.1105 (0.0869) | 0.1087 (0.0858) | 0.1706 (0.1002) | 0.1082 (0.0837) | 0.1055 (0.0829) |
| | III | $v$ | 0.0980 (0.0751) | 0.0629 (0.0418) | 0.0606 (0.0411) | 0.0954 (0.0739) | 0.0609 (0.0398) | 0.0583 (0.0395) |
| | | $\eta$ | 0.0896 (0.0705) | 0.0595 (0.0369) | 0.0582 (0.0365) | 0.0875 (0.0699) | 0.0589 (0.0363) | 0.0562 (0.0363) |
| | | $S(0.25)$ | 0.0122 (0.0090) | 0.0086 (0.0062) | 0.0085 (0.0061) | 0.0115 (0.0088) | 0.0084 (0.0060) | 0.0083 (0.0060) |
| | | $h(0.25)$ | 0.1589 (0.0994) | 0.1016 (0.0798) | 0.1001 (0.0791) | 0.1523 (0.0941) | 0.0982 (0.0791) | 0.0967 (0.0785) |
| (80,40) | I | $v$ | 0.0840 (0.0661) | 0.0473 (0.0319) | 0.0458 (0.0304) | 0.0815 (0.0635) | 0.0458 (0.0306) | 0.0441 (0.0301) |
| | | $\eta$ | 0.0793 (0.0602) | 0.0452 (0.0308) | 0.0443 (0.0299) | 0.0764 (0.0590) | 0.0441 (0.0302) | 0.0438 (0.0297) |
| | | $S(0.25)$ | 0.0115 (0.0088) | 0.0082 (0.0056) | 0.0080 (0.0055) | 0.0110 (0.0083) | 0.0080 (0.0055) | 0.0078 (0.0054) |
| | | $h(0.25)$ | 0.1475 (0.0945) | 0.0956 (0.0794) | 0.0947 (0.0789) | 0.1439 (0.0922) | 0.0941 (0.0781) | 0.0932 (0.0772) |
| | II | $v$ | 0.0875 (0.0686) | 0.0502 (0.0344) | 0.0484 (0.0327) | 0.0859 (0.0667) | 0.0485 (0.0327) | 0.0480 (0.0324) |
| | | $\eta$ | 0.0822 (0.0631) | 0.0474 (0.0319) | 0.0461 (0.0312) | 0.0789 (0.0610) | 0.0455 (0.0313) | 0.0447 (0.0308) |
| | | $S(0.25)$ | 0.0120 (0.0089) | 0.0084 (0.0057) | 0.0082 (0.0056) | 0.0116 (0.0085) | 0.0082 (0.0056) | 0.0080 (0.0055) |
| | | $h(0.25)$ | 0.1511 (0.0963) | 0.0967 (0.0801) | 0.0955 (0.0797) | 0.1460 (0.0939) | 0.0952 (0.0794) | 0.0941 (0.0783) |
| | III | $v$ | 0.0831 (0.0652) | 0.0464 (0.0315) | 0.0453 (0.0301) | 0.0802 (0.0619) | 0.0451 (0.0299) | 0.0427 (0.0295) |
| | | $\eta$ | 0.0778 (0.0596) | 0.0447 (0.0302) | 0.0438 (0.0296) | 0.0753 (0.0584) | 0.0432 (0.0298) | 0.0424 (0.0294) |
| | | $S(0.25)$ | 0.0111 (0.0085) | 0.0080 (0.0054) | 0.0078 (0.0053) | 0.0104 (0.0080) | 0.0077 (0.0054) | 0.0075 (0.0053) |
| | | $h(0.25)$ | 0.1434 (0.0929) | 0.0934 (0.0762) | 0.0929 (0.0759) | 0.1418 (0.0910) | 0.0928 (0.0758) | 0.0923 (0.0755) |

(*Continued*)

**Table 2.** (Continued)

| (n, m) | CS | | $T_1 = 0.4, T_2 = 0.8$ | | | $T_1 = 0.55, T_2 = 1.1$ | | |
|---|---|---|---|---|---|---|---|---|
| | | | **MLE** | **SELF** | **LLF** | **MLE** | **SELF** | **LLF** |
| (80,60) | I | $v$ | 0.0788 (0.0637) | 0.0458 (0.0304) | 0.0449 (0.0296) | 0.0774 (0.0625) | 0.0447 (0.0302) | 0.0435 (0.0298) |
| | | $\eta$ | 0.0768 (0.0587) | 0.0443 (0.0296) | 0.0439 (0.0294) | 0.0747 (0.0572) | 0.0438 (0.0294) | 0.0434 (0.0291) |
| | | $S(0.25)$ | 0.0104 (0.0082) | 0.0076 (0.0051) | 0.0074 (0.0050) | 0.0099 (0.0080) | 0.0075 (0.0050) | 0.0074 (0.0050) |
| | | $h(0.25)$ | 0.1418 (0.0919) | 0.0922 (0.0770) | 0.0916 (0.0764) | 0.1397 (0.0904) | 0.0915 (0.0763) | 0.0907 (0.0754) |
| | II | $v$ | 0.0841 (0.0675) | 0.0489 (0.0331) | 0.0477 (0.0320) | 0.0834 (0.0656) | 0.0472 (0.0318) | 0.0467 (0.0316) |
| | | $\eta$ | 0.0796 (0.0618) | 0.0458 (0.0304) | 0.0449 (0.0301) | 0.0767 (0.0597) | 0.0450 (0.0301) | 0.0441 (0.0297) |
| | | $S(0.25)$ | 0.0108 (0.0084) | 0.0079 (0.0054) | 0.0077 (0.0053) | 0.0104 (0.0080) | 0.0076 (0.0052) | 0.0075 (0.0051) |
| | | $h(0.25)$ | 0.1483 (0.0941) | 0.0955 (0.0792) | 0.0946 (0.0790) | 0.1421 (0.0927) | 0.0945 (0.0783) | 0.0939 (0.0778) |
| | III | $v$ | 0.0820 (0.0646) | 0.0459 (0.0312) | 0.0451 (0.0297) | 0.0795 (0.0611) | 0.0442 (0.0293) | 0.0416 (0.0291) |
| | | $\eta$ | 0.0745 (0.0572) | 0.0436 (0.0289) | 0.0430 (0.0285) | 0.0731 (0.0558) | 0.0430 (0.0286) | 0.0427 (0.0284) |
| | | $S(0.25)$ | 0.0099 (0.0080) | 0.0074 (0.0049) | 0.0072 (0.0048) | 0.0095 (0.0079) | 0.0072 (0.0048) | 0.0070 (0.0047) |
| | | $h(0.25)$ | 0.1359 (0.0895) | 0.0904 (0.0761) | 0.0901 (0.0760) | 0.1342 (0.0891) | 0.0896 (0.0758) | 0.0894 (0.0751) |

These findings demonstrate that the proposed estimation methods provide robust and precise results for increasing values of *n*, *m*, and $(T_1, T_2)$. The decrease in ABs and MSEs indicates improved precision of the point estimates, while the consistent CPs across varying sample sizes and censoring schemes imply that the proposed interval estimates adequately represent the true parameter values. One can easily summarize that in case of point estimation, BEs under LLF performs better than the other proposed estimates and for interval estimation, HPD credible interval outperforms the ACIs.

## 6 Mechanical equipment data analysis

An engineering application employing a real-life data set reported by Murthy et al. [24] is analyzed to demonstrate how approaches suggested may be tailored to real phenomena. In recent time, Elshahhat et al. [25] reanalyzed this RME data set for AT-II PCS following weighted-exponential distribution. This data set which demonstrates the intervals between failures for thirty pieces of repairable mechanical equipment (RME) is given below:

0.11, 0.30, 0.40, 0.45, 0.59, 0.63, 0.70, 0.71, 0.74, 0.77, 0.94, 1.06, 1.17, 1.23, 1.23, 1.24,

1.43, 1.46, 1.49, 1.74, 1.82, 1.86, 1.97, 2.23, 2.37, 2.46, 2.63, 3.46, 4.36, 4.73.

Before proceeding with the analysis, it is important to determine whether the proposed model adequately fits the data. To achieve this, the MLEs of *v* and *η* based on the real data set were determined to be 1.5864 and 0.5415, respectively. In addition, the corresponding Kolmogorov-Smirnov (K-S) distance and *p*-value were obtained as 0.0732 and 0.9933, respectively. To further assess the goodness-of-fit, Fig 3 displays the empirical CDF (ECDF), probability-probability (P-P), and quantile-quantile (Q-Q) plots for the complete data set. Furthermore, the Weibull distribution were examined as a potential alternative model, represented by the CDF, $F(x) = 1 - \exp(-(x/\sigma)^v)$, where $x > 0$, and $v, \sigma > 0$ represent the shape and scale parameters, respectively. Based on RME data, the K-S distance and corresponding *p*-value for the Weibull distribution are 0.0748 and 0.9914, respectively. These results demonstrate that the LED fits the data set quite well.

**Table 3. AWs and CPs (in parenthesis) of the 95% interval estimates of $v$, $\eta$, S(0.5), and h(0.5) with different values of $n$, $m$, and ($T_1$, $T_2$).**

| (n, m) | CS | | $T_1 = 0.4$, $T_2 = 0.8$ | | $T_1 = 0.55$, $T_2 = 1.1$ | |
|---|---|---|---|---|---|---|
| | | | **ACI** | **HPD** | **ACI** | **HPD** |
| (40,20) | I | $v$ | 0.8498 (0.8961) | 0.4869 (0.9278) | 0.8051 (0.9075) | 0.4550 (0.9324) |
| | | $\eta$ | 0.7781 (0.9014) | 0.4650 (0.9305) | 0.7355 (0.9108) | 0.4498 (0.9371) |
| | | S(0.25) | 0.6528 (0.8795) | 0.3991 (0.9250) | 0.6384 (0.8834) | 0.3897 (0.9345) |
| | | h(0.25) | 1.0249 (0.9047) | 0.6152 (0.9341) | 0.9563 (0.9142) | 0.5882 (0.9413) |
| | II | $v$ | 0.8635 (0.8756) | 0.4967 (0.9065) | 0.8190 (0.8934) | 0.4718 (0.9176) |
| | | $\eta$ | 0.7922 (0.8867) | 0.4783 (0.9165) | 0.7486 (0.9044) | 0.4603 (0.9165) |
| | | S(0.25) | 0.6743 (0.8640) | 0.4125 (0.9041) | 0.6534 (0.8769) | 0.4034 (0.9162) |
| | | h(0.25) | 1.0845 (0.8890) | 0.6324 (0.9167) | 1.0121 (0.9033) | 0.5974 (0.9265) |
| | III | $v$ | 0.8276 (0.9045) | 0.4755 (0.9345) | 0.8134 (0.9108) | 0.4474 (0.9351) |
| | | $\eta$ | 0.7565 (0.9065) | 0.4724 (0.9351) | 0.7406 (0.9143) | 0.4563 (0.9398) |
| | | S(0.25) | 0.6395 (0.8874) | 0.3937 (0.9304) | 0.6299 (0.8975) | 0.3816 (0.9381) |
| | | h(0.25) | 0.9851 (0.9162) | 0.6033 (0.9402) | 0.9372 (0.9299) | 0.5750 (0.9436) |
| (40,30) | I | $v$ | 0.8095 (0.9133) | 0.4742 (0.9336) | 0.7834 (0.9188) | 0.4434 (0.9395) |
| | | $\eta$ | 0.6885 (0.9384) | 0.4275 (0.9435) | 0.6769 (0.9279) | 0.4065 (0.9467) |
| | | S(0.25) | 0.6359 (0.8914) | 0.3850 (0.9367) | 0.6206 (0.8972) | 0.3765 (0.9392) |
| | | h(0.25) | 0.9582 (0.9159) | 0.5984 (0.9396) | 0.9332 (0.9265) | 0.5691 (0.9430) |
| | II | $v$ | 0.8246 (0.9074) | 0.4865 (0.9263) | 0.8046 (0.9136) | 0.4567 (0.9315) |
| | | $\eta$ | 0.7486 (0.9034) | 0.4351 (0.9271) | 0.7137 (0.9186) | 0.4485 (0.9293) |
| | | S(0.25) | 0.6473 (0.8875) | 0.4042 (0.9153) | 0.6381 (0.8867) | 0.3890 (0.9245) |
| | | h(0.25) | 0.9634 (0.9076) | 0.6049 (0.9335) | 0.9406 (0.9171) | 0.5723 (0.9372) |
| | III | $v$ | 0.7862 (0.9207) | 0.4593 (0.9370) | 0.7715 (0.9263) | 0.4338 (0.9411) |
| | | $\eta$ | 0.7408 (0.9143) | 0.4649 (0.9398) | 0.7251 (0.9261) | 0.4482 (0.9423) |
| | | S(0.25) | 0.6174 (0.9035) | 0.3881 (0.9383) | 0.6074 (0.9034) | 0.3674 (0.9423) |
| | | h(0.25) | 0.9570 (0.9249) | 0.5882 (0.9435) | 0.9155 (0.9346) | 0.5626 (0.9450) |
| (80,40) | I | $v$ | 0.7539 (0.9250) | 0.4552 (0.9383) | 0.7420 (0.9296) | 0.4341 (0.9417) |
| | | $\eta$ | 0.6453 (0.9429) | 0.3947 (0.9446) | 0.6435 (0.9351) | 0.3892 (0.9477) |
| | | S(0.25) | 0.5921 (0.9083) | 0.3688 (0.9414) | 0.5849 (0.9127) | 0.3621 (0.9429) |
| | | h(0.25) | 0.9258 (0.9274) | 0.5686 (0.9431) | 0.9185 (0.9304) | 0.5601 (0.9442) |
| | II | $v$ | 0.7959 (0.9123) | 0.4642 (0.9336) | 0.7750 (0.9204) | 0.4477 (0.9349) |
| | | $\eta$ | 0.7095 (0.9095) | 0.4183 (0.9326) | 0.6890 (0.9239) | 0.4141 (0.9361) |
| | | S(0.25) | 0.6272 (0.8980) | 0.3956 (0.9218) | 0.6164 (0.8958) | 0.3750 (0.9339) |
| | | h(0.25) | 0.9587 (0.9157) | 0.5952 (0.9374) | 0.9276 (0.9250) | 0.5591 (0.9399) |
| | III | $v$ | 0.7453 (0.9294) | 0.4517 (0.9406) | 0.7378 (0.9329) | 0.4282 (0.9435) |
| | | $\eta$ | 0.6381 (0.9438) | 0.3899 (0.9458) | 0.6273 (0.9389) | 0.3834 (0.9485) |
| | | S(0.25) | 0.5814 (0.9156) | 0.3593 (0.9432) | 0.5675 (0.9206) | 0.3579 (0.9441) |
| | | h(0.25) | 0.8959 (0.9361) | 0.5472 (0.9444) | 0.9264 (0.9351) | 0.5465 (0.9463) |

(*Continued*)

**Table 3.** (Continued)

| (n, m) | CS | | T₁ = 0.4, T₂ = 0.8 | | T₁ = 0.55, T₂ = 1.1 | |
|---|---|---|---|---|---|---|
| | | | ACI | HPD | ACI | HPD |
| (80,60) | I | $v$ | 0.7287 (0.9312) | 0.4472 (0.9414) | 0.7266 (0.9347) | 0.4259 (0.9435) |
| | | $\eta$ | 0.6289 (0.9443) | 0.3865 (0.9460) | 0.6280 (0.9384) | 0.3761 (0.9481) |
| | | $S(0.25)$ | 0.5786 (0.9154) | 0.3599 (0.9423) | 0.5750 (0.9188) | 0.3582 (0.9440) |
| | | $h(0.25)$ | 0.8955 (0.9304) | 0.5571 (0.9442) | 0.8843 (0.9357) | 0.5523 (0.9449) |
| | II | $v$ | 0.7665 (0.9184) | 0.4592 (0.9379) | 0.7580 (0.9267) | 0.4403 (0.9391) |
| | | $\eta$ | 0.6156 (0.9449) | 0.3763 (0.9471) | 0.6089 (0.9425) | 0.3772 (0.9489) |
| | | $S(0.25)$ | 0.6082 (0.9063) | 0.3894 (0.9259) | 0.6025 (0.9120) | 0.3689 (0.9383) |
| | | $h(0.25)$ | 0.9405 (0.9212) | 0.5876 (0.9409) | 0.9085 (0.9327) | 0.5468 (0.9419) |
| | III | $v$ | 0.7286 (0.9345) | 0.4439 (0.9431) | 0.7199 (0.9365) | 0.4153 (0.9449) |
| | | $\eta$ | 0.6240 (0.9449) | 0.3782 (0.9471) | 0.6054 (0.9414) | 0.3696 (0.9488) |
| | | $S(0.25)$ | 0.5529 (0.9283) | 0.3508 (0.9455) | 0.5359 (0.9278) | 0.3488 (0.9455) |
| | | $h(0.25)$ | 0.8782 (0.9404) | 0.5336 (0.9458) | 0.9091 (0.9392) | 0.5373 (0.9481) |

Now, three IAT-II PCS samples are generated from the complete data set, with $m$ = 14 and different choices of $(R_1, \cdots, R_m)$, $T_1$, and $T_2$. Table 3 reports the produced samples together with the relevant censoring schemes.

The point and interval estimates for the parameters $v$ and $\eta$ and the reliability characteristics $S(t)$ and $h(t)$ at $t$ = 0.5 have been obtained by using the generated IAT-II PCS samples which are tabulated in Table 4. To show the existence of uniqueness of the MLEs of the parameters in graphical approach, a contour plot is depicted in Fig 4. To obtain the Bayes estimates, 5, 000 MCMC samples have been generated and gamma priors have been considered with hyperparameters $a = b = c = d$ = 0.001. All these point and interval estimates based on the RME data have been reported in Table 5. To check the convergence of the MCMC samples, trace plots are given here in Fig 5. From Table 5, it has been concluded that Bayes estimates

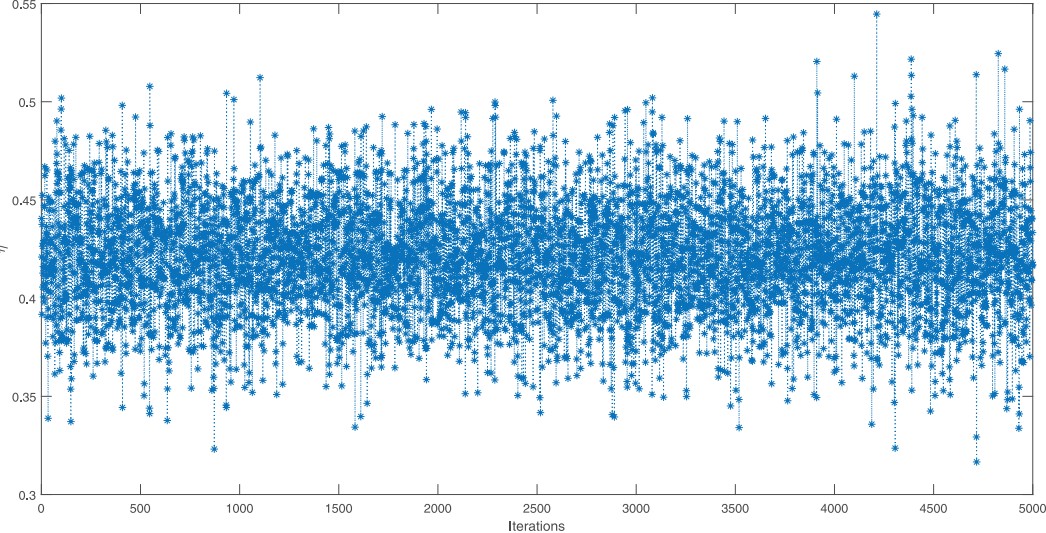

**Fig 3. ECDF, P-P, and Q-Q plots for the LED distribution with real data.**

**Table 4. Three different IAT-II PCS samples generated from RME data.**

| Sample | PCS | $T_1$ | $T_2$ | IAT-II PCS | $E_1$ | $E_2$ | $R^*$ | $T^*$ |
|---|---|---|---|---|---|---|---|---|
| A | (4 * 2, 0 *10, 4 * 2) | 1.25 | 2.25 | 0.30, 0.45, 0.59, 0.70, 0.77, 0.94, 1.06, 1.17, 1.43, 1.74, 1.82, 1.86, 2.23 | 8 | 13 | 9 | 2.25 |
| B | (0 * 6, 2 * 8) | 1.5 | 2.5 | 0.11, 0.40, 0.59, 0.74, 0.94, 1.06, 1.23, 1.49, 1.74, 1.82, 1.97, 2.23, 2.37, 2.46 | 8 | 14 | 12 | 2.46 |
| C | (2 * 4, 0 * 6, 2 * 4) | 1.0 | 2.0 | 0.11, 0.30, 0.40, 0.59, 0.74, 0.94, 1.06, 1.23, 1.49, 1.74, 1.82, 1.97 | 6 | 12 | 10 | 2.0 |

perform better than MLEs in terms of the standard error (SE). It has been observed that HPD credible intervals perform better than ACIs with 5% significance level.

## 7 Conclusions

This article examines the reliability, hazard rate functions, and unknown parameter estimation issues of the logistic-exponential distribution using an IAT-II PCS. The maximum likelihood estimates and the associated asymptotic confidence intervals have been obtained. The Bayes estimates are derived based on gamma priors and the corresponding highest posterior density credible intervals are also constructed. To obtain the Bayes estimates, the MCMC technique has been employed. A Monte Carlo simulation study is carried out under various circumstances to compare the behavior of the different estimations. In terms of average length, coverage probability, average bias, and mean square error, the Bayesian estimates outperformed the classical estimates. Eventually, a real-life data set has been analyzed to demonstrate the suitability of the suggested techniques. In case of simulation studies, when we choose small number of experimental units (less than 10), the results are not found as usual. This type of limitation has been found in this study. As a future direction, one may consider the various approaches

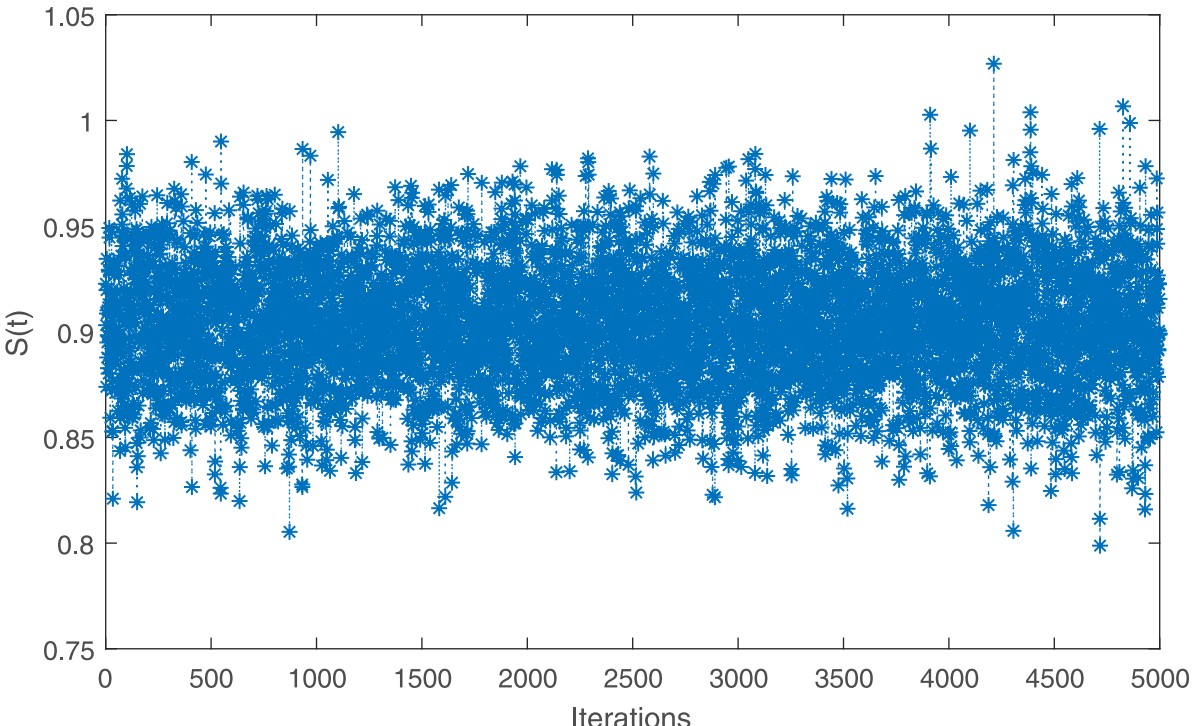

**Fig 4. Contour plot of the log-likelihood function based on the IAT-II PCS sample 'C' using the RME data.**

**Table 5. Point estimates (Standard errors in parenthesis) and interval estimates (interval lengths in parenthesis) for these three IAT-II PCS samples generated from RME data.**

| Sample | | MLE | Bayes | ACI | HPD |
|---|---|---|---|---|---|
| A | $v$ | 1.5620 (0.1112) | 1.6021 (0.0784) | (0.9084, 2.2155) [1.3071] | (1.2519, 1.9820) [0.7301] |
| | $\eta$ | 0.3664 (0.0129) | 0.4219 (0.0082) | (0.2362, 0.4967) [0.2605] | (0.3743, 0.4697) [0.0954] |
| | $S(0.5)$ | 0.9245 (0.0067) | 0.9041 (0.0031) | (0.7991, 1.0499) [0.2508] | (0.8653, 0.9448) [0.0795] |
| | $h(0.5)$ | 0.2580 (0.0934) | 0.3653 (0.0141) | (0.0569, 0.4591) [0.4022] | (0.2899, 0.4246) [0.1367] |
| B | $v$ | 1.4202 (0.0947) | 1.5349 (0.0296) | (0.9570, 1.8836) [0.9266] | (1.3244, 1.7517) [0.4273] |
| | $\eta$ | 0.2940 (0.0072) | 0.4841 (0.0034) | (0.1847, 0.4032) [0.2185] | (0.4439, 0.5187) [0.0748] |
| | $S(0.5)$ | 0.9319 (0.0058) | 0.9025 (0.0025) | (0.8328, 1.0310) [0.1982] | (0.8718, 0.9375) [0.0657] |
| | $h(0.5)$ | 0.2077 (0.1145) | 0.3362 (0.0321) | (0.0247, 0.3907) [0.3660] | (0.2565, 0.4294) [0.1729] |
| C | $v$ | 1.3406 (0.0930) | 1.5419 (0.0214) | (0.7428, 1.9383) [1.1955] | (1.3785, 1.7209) [0.3424] |
| | $\eta$ | 0.3655 (0.0064) | 0.4952 (0.0029) | (0.2089, 0.5221) [0.3132] | (0.4637, 0.5394) [0.0757] |
| | $S(0.5)$ | 0.8960 (0.0050) | 0.8791 (0.0029) | (0.7574, 1.0346) [0.2772] | (0.8408, 0.9141) [0.0733] |
| | $h(0.5)$ | 0.3049 (0.1860) | 0.3768 (0.0416) | (0.1404, 0.5018) [0.3614] | (0.2817, 0.4589) [0.1772] |

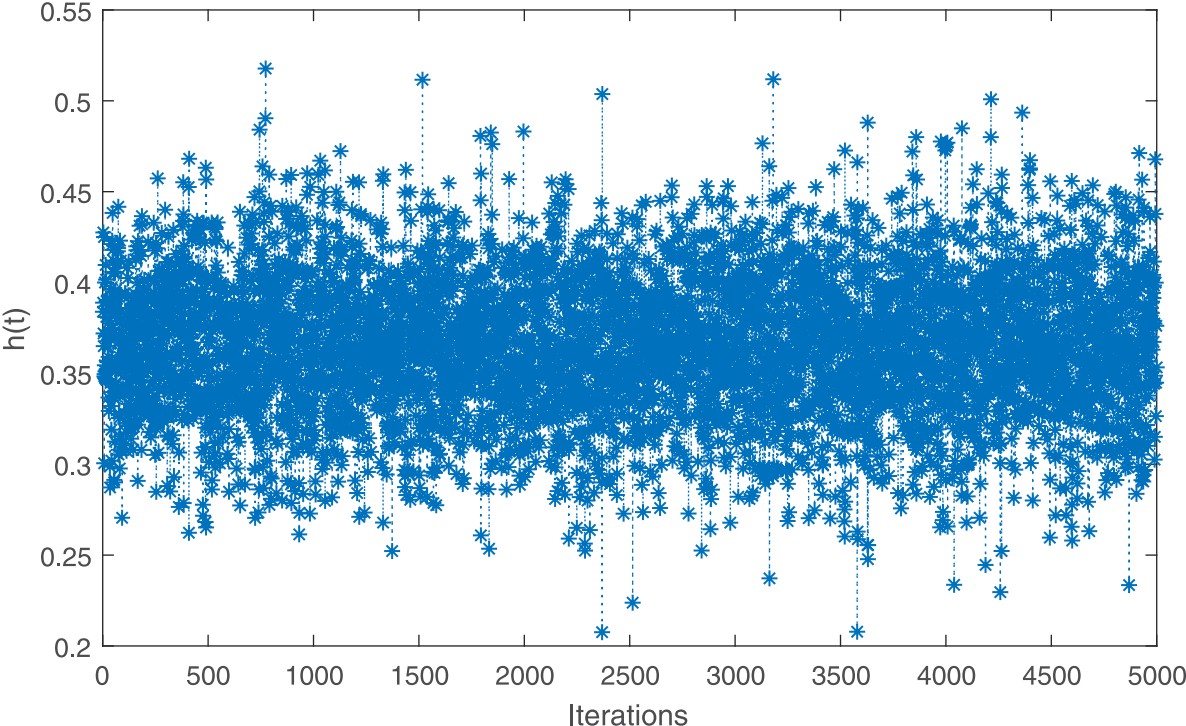

**Fig 5. Trcae plots of the MCMC samples of $v$, $\eta$, $S(t)$, and $h(t)$ for IAT-II PCS sample A generated from the real data.**

covered in the study in the context of accelerated life testing model for different lifetime distributions would be fascinating. The work is ongoing and will be reported at a later time.

## Acknowledgments

Researchers would like to thank the Deanship of Scientific Research, Qassim University for funding publication of this project.

## Author Contributions

**Conceptualization:** Subhankar Dutta, Hana N. Alqifari, Amani Almohaimeed.

**Data curation:** Subhankar Dutta.

**Formal analysis:** Subhankar Dutta.

**Funding acquisition:** Subhankar Dutta.

**Investigation:** Subhankar Dutta, Amani Almohaimeed.

**Methodology:** Hana N. Alqifari, Amani Almohaimeed.

**Project administration:** Hana N. Alqifari.

**Resources:** Hana N. Alqifari, Amani Almohaimeed.

**Supervision:** Subhankar Dutta.

**Validation:** Subhankar Dutta.

**Visualization:** Subhankar Dutta.

**Writing – original draft:** Subhankar Dutta, Hana N. Alqifari, Amani Almohaimeed.

**Writing – review & editing:** Subhankar Dutta, Hana N. Alqifari, Amani Almohaimeed.

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
