## [Decision Letter · Decision Letter 0]

18 Dec 2023

PONE-D-23-40731Bayesian and Non-Bayesian inference for Logistic-exponential Distribution using Improved Adoptive Type-II Progressively Censored Data.PLOS ONE

Dear Dr. Alqifari,

Thank you for submitting your manuscript to PLOS ONE. After careful consideration, we feel that your manuscript will likely be suitable for publication if it is revised to address the points below. Therefore, my decision is "Minor Revision".

We look forward to receiving your revised manuscript.

Kind regards,

Oluwafemi Samson Balogun, Ph.D.

Academic Editor

PLOS ONE

Journal Requirements:

4. Please upload a copy of Figures 1-3, to which you refer in your text on pages 3 and 14. If the figure is no longer to be included as part of the submission please remove all reference to it within the text.

Reviewers' comments:

Reviewer's Responses to Questions

**Comments to the Author**

1. Is the manuscript technically sound, and do the data support the conclusions?

Reviewer #1: Yes

Reviewer #2: Yes

2. Has the statistical analysis been performed appropriately and rigorously? 

Reviewer #1: Yes

Reviewer #2: Yes

3. Have the authors made all data underlying the findings in their manuscript fully available?

Reviewer #1: Yes

Reviewer #2: Yes

4. Is the manuscript presented in an intelligible fashion and written in standard English?

Reviewer #1: Yes

Reviewer #2: Yes

5. Review Comments to the Author

Reviewer #1: Report on: “Bayesian and Non-Bayesian inference for Logistic-exponential Distribution using Improved Adoptive Type-II Progressively Censored Data.”

Overall, I find the paper very interesting, but I think there is some room for improvement. Please, find below some comments:

1. The literature review of the different improved adoptive type-II progressively censored data is very good. I commend the authors for presenting a balanced account of the literature on adoptive type-II progressively censored data.

2. Please, indicate the motivation behind using the mechanical equipment data

3. It is not necessary to enumerate all equations. Only the ones to be referred to.

4. A word of caution regarding the introduction section, I commend the authors to split the introduction section into two different sections, section I entitled “Introduction” and another section entitled “model formulation” to avoid misguiding

5. The baseline model proposed in this study is logistic-exponential. I strongly recommend discussing about what motivates to use this baseline compared to other existing ones.

6. In general, I encourage the authors to reflect on whether the included material goes beyond a marginal contribution that perhaps could be find on textbooks instead. This will help them make a more careful selection of the material included in the paper, and to present a more concise message.

7. Overall, I think the paper could be shorten to about 15 pages without affecting the main contributions.

8. A detailed subsection about the convergence diagnostics is recommendable

9. A proofread is recommended.

Reviewer #2: The paper can be accepted after solving the following issues.

1- What were the main findings or insights you gained from the numerical analysis? Were there any unexpected or counterintuitive results? 2-How did you select the two actual data sets for demonstration purposes? 3-What were the characteristics of these data sets, and how did they relate to the theoretical results of the study?

4-What are some future directions for research in this area?

5-Are there any other estimation methods or entropy measures that could be explored?

6. How can the confidence intervals of the entropies be estimated using various estimation methods?

7- There is a need of some more literature regarding the Adaptive Type-II Progressive Censored and the different estimation procedures of distribution. The author may use some references such as:

Bayesian and non-Bayesian inference for inverse Weibull model based on jointly type-II hybrid censoring samples with modeling to physics data

Analysis of WE Parameters of Life Using Adaptive-Progressively Type-II Hybrid Censored Mechanical Equipment Data

Optimal Test Plan of Step Stress Partially Accelerated Life Testing for Alpha Power Inverse Weibull Distribution under Adaptive Progressive Hybrid Censored Data and Different Loss Functions

Inferential Survival Analysis for Inverted NH Distribution Under Adaptive Progressive Hybrid Censoring with Application of Transformer Insulation

Maximum Product Spacing Estimation of Weibull Distribution Under Adaptive Type-II Progressive Censoring Schemes

Optimal Analysis of Adaptive Type-II Progressive Censored for New Unit-Lindley Model

Data Analysis by Adaptive Progressive Hybrid Censored Under Bivariate Model

Inferential Survival Analysis for Inverted NH Distribution Under Adaptive Progressive Hybrid Censoring with Application of Transformer Insulation

Statistical Inference for the Extended Weibull Distribution Based on Adaptive Type-II Progressive Hybrid Censored Competing Risks Data

Bayesian and non-Bayesian inference under adaptive type-II progressive censored sample with exponentiated power Lindley distribution

The Weibull Generalized Exponential Distribution with Censored Sample: Estimation and Application on Real Data

8. Discuss the behavior of graphs in detail.

9. How do we distinguish that the parameters of MLE are not a local maximum?

10. The authors should explain how the p-values for the KS test have been calculated. This is not a straightforward problem in situations when model parameters have to be estimated.

11. The authors use MCMC to estimate the posterior density function. However, there are other estimators such as KDE with tighter convergence rates that they could explore.

12. Take care of the formatting of the references.

13. In the references section, it is necessary to remove the reference that the author doesn't use in the manuscript. Please check.

6. PLOS authors have the option to publish the peer review history of their article (what does this mean?). If published, this will include your full peer review and any attached files.

Reviewer #1: No

Reviewer #2: No

---

## [Author Response · Author response to Decision Letter 0]

27 Dec 2023

We sincerely appreciate the reviewers insightful comments and the detailed suggestions for improving the earlier version of this manuscript.

---

## [Decision Letter · Decision Letter 1]

8 Jan 2024

PONE-D-23-40731R1Bayesian and Non-Bayesian inference for Logistic-exponential Distribution using Improved Adoptive Type-II Progressively Censored Data.PLOS ONE

Dear Dr. Hana Nasser Alqifari,

Thank you for submitting your manuscript to PLOS ONE. After careful consideration, we feel that your manuscript will likely be suitable for publication if it is revised to address the points below. Therefore, my decision is "Minor Revision".

Please submit your revised manuscript by Feb 22 2024 11:59PM If you will need more time than this to complete your revisions, please reply to this message or contact the journal office at plosone@plos.org. Please include the following items when submitting your revised manuscript:A rebuttal letter that responds to each point raised by the academic editor and reviewer(s). You should upload this letter as a separate file labeled 'Response to Reviewers'.A marked-up copy of your manuscript that highlights changes made to the original version. You should upload this as a separate file labeled 'Revised Manuscript with Track Changes'.An unmarked version of your revised paper without tracked changes. You should upload this as a separate file labeled 'Manuscript'.If applicable, we recommend that you deposit your laboratory protocols in protocols.io to enhance the reproducibility of your results. Protocols.io assigns your protocol its own identifier (DOI) so that it can be cited independently in the future. For instructions see: https://journals.plos.org/plosone/s/submission-guidelines#loc-laboratory-protocols. Additionally, PLOS ONE offers an option for publishing peer-reviewed Lab Protocol articles, which describe protocols hosted on protocols.io. Read more information on sharing protocols at https://plos.org/protocols?utm_medium=editorial-email&utm_source=authorletters&utm_campaign=protocols.

We look forward to receiving your revised manuscript.

Kind regards,

Oluwafemi Samson Balogun, Ph.D.

Academic Editor

PLOS ONE

Journal Requirements:

Reviewers' comments:

Reviewer's Responses to Questions

**Comments to the Author**

1. If the authors have adequately addressed your comments raised in a previous round of review and you feel that this manuscript is now acceptable for publication, you may indicate that here to bypass the “Comments to the Author” section, enter your conflict of interest statement in the “Confidential to Editor” section, and submit your "Accept" recommendation.

Reviewer #1: All comments have been addressed

Reviewer #3: (No Response)

2. Is the manuscript technically sound, and do the data support the conclusions?

Reviewer #1: Yes

Reviewer #3: Partly

3. Has the statistical analysis been performed appropriately and rigorously? 

Reviewer #1: Yes

Reviewer #3: Yes

4. Have the authors made all data underlying the findings in their manuscript fully available?

Reviewer #1: Yes

Reviewer #3: Yes

5. Is the manuscript presented in an intelligible fashion and written in standard English?

Reviewer #1: Yes

Reviewer #3: (No Response)

6. Review Comments to the Author

Reviewer #1: (No Response)

Reviewer #3: 1. In section 1, the authors used the Logistic-exponential distribution as a lifetime distribution, please write the authors who introduced this distribution. Define the type of each parameter. Add plots of the density and hazard rate functions according to the descriptions mentioned after equation 1.4.

2. In section 1, at the end of second paragraph, this sentence “This work specifically examines censored samples using an enhanced adaptive Type-II progressive censoring method. The objective is to estimate the unknown parameters of the LED” has no meaning there, please remove it.

3. In log-likelihood function equation this term must be corrected to , and must be corrected in equation (3.5) to

4. Also this term must be corrected to, also this term must be corrected in all subsequent equations (3.3), (3.4),(3.5).

5. According to the previous corrections all the second derivatives must be checked carefully? Also, check all equations carefully.

6. In equation (3.7), must be corrected to Also, the same correction must be made for the observed FIM.

7. At the end of page 7, approximate confidence intervals (ACIs) must be written as ACIs

8. After equation (4.1), SELF, and LLF must be defined. After algorithm 1 SEL and LL must be defined.

9. The selected value of p in linear exponential loss function must be added in section 5.

10. The visibility and clarity of all tables must be enhanced. In table 5, (SEs in parenthesis), SEs must be defined

11. Why do the authors consider just this only one true value of parameter in simulation study? Do they correspond to any experience from practice?

12. The conclusions should be added by discussing limitations of the proposed methods?

Some corrections

13. Cohen and Clifford [1] must be corrected to Cohen [1]

14. In introduction, this abbreviation PDF and CDF must be defined.

15. In introduction section, LE distribution must be written as LED, third line of last paragraph in introduction section LE replaced with LED. In all article check the abbreviation LE?

16. At the end of page 3, the authors write pdf and cdf, write this abbreviation in only one form, pdf, cdf or PDF, CDF.

7. PLOS authors have the option to publish the peer review history of their article (what does this mean?). If published, this will include your full peer review and any attached files.

Reviewer #1: No

Reviewer #3: No

---

## [Author Response · Author response to Decision Letter 1]

17 Jan 2024

We express our sincere gratitude to the Reviewers for their valuable feedback, which significantly enhanced our manuscript.

---

## [Decision Letter · Decision Letter 2]

29 Jan 2024

Bayesian and Non-Bayesian inference for Logistic-exponential Distribution using Improved Adaptive Type-II Progressively Censored Data

PONE-D-23-40731R2

Dear Dr. Hana Nasser Alqifari,

We’re pleased to inform you that your manuscript has been judged scientifically suitable for publication and will be formally accepted for publication once it meets all outstanding technical requirements.

Kind regards,

Oluwafemi Samson Balogun, Ph.D.

Academic Editor

PLOS ONE

Additional Editor Comments (optional):

Reviewers' comments:

Reviewer's Responses to Questions

**Comments to the Author**

1. If the authors have adequately addressed your comments raised in a previous round of review and you feel that this manuscript is now acceptable for publication, you may indicate that here to bypass the “Comments to the Author” section, enter your conflict of interest statement in the “Confidential to Editor” section, and submit your "Accept" recommendation.

Reviewer #1: All comments have been addressed

Reviewer #3: All comments have been addressed

2. Is the manuscript technically sound, and do the data support the conclusions?

Reviewer #1: Yes

Reviewer #3: Yes

3. Has the statistical analysis been performed appropriately and rigorously? 

Reviewer #1: Yes

Reviewer #3: Yes

4. Have the authors made all data underlying the findings in their manuscript fully available?

Reviewer #1: Yes

Reviewer #3: Yes

5. Is the manuscript presented in an intelligible fashion and written in standard English?

Reviewer #1: Yes

Reviewer #3: Yes

6. Review Comments to the Author

Reviewer #1: The paper is revised as per request and now deserves to accept in its current form. Authors have answered and responded all the comments we raised

Reviewer #3: (No Response)

7. PLOS authors have the option to publish the peer review history of their article (what does this mean?). If published, this will include your full peer review and any attached files.

Reviewer #1: No

Reviewer #3: No

---

## [Editor Report · Acceptance letter]

15 Feb 2024

PONE-D-23-40731R2 

PLOS ONE

Dear Dr. Alqifari, 

I'm pleased to inform you that your manuscript has been deemed suitable for publication in PLOS ONE. Congratulations! Your manuscript is now being handed over to our production team.

Kind regards, 

on behalf of

Dr. Oluwafemi Samson Balogun 

Academic Editor

PLOS ONE